# Diagnosis of *Helicobacter pylori* infection: serology vs. urea breath test

Miguel Imperial,[1,2,3] Kennard Tan,[1,4] Chris Fjell,[5] Yin Chang,[5] Mel Krajden,[1,5] Michael T. Kelly,[1,3] Muhammad Morshed[1,5]

**ABSTRACT** The objective of the study was to ascertain an optimal *Helicobacter pylori* diagnostic strategy using population-level laboratory data comparing the performance of serology against urea breath test (UBT). *H. pylori* diagnostic test results for serology and UBT from two laboratories over a 12-year period (2006–20017) were extracted, linked, and analyzed. A subset of this population underwent both methods of testing within days of each other, enabling a direct comparison of the two methods. The average prevalence of *H. pylor*i positivity was 21.3% by serology and 17.5% by UBT. There were 2,612 individuals who had serology performed first, followed by UBT within 14 days. For this subset, the sensitivity of serology compared with UBT was 96.5% with a specificity of 79.2%. The negative predictive value for serology was 98.4%. Contrary to various recent clinical guidelines, the data show that serology still has utility as a sensitive enough test to be used as an initial *H. pylori* screening test in a lower prevalence population. Negative serology can be used with confidence to rule out active infection, whereas a positive serology could be followed up with a UBT or a similar performing test such as stool antigen to differentiate active from past infection. For population-based diagnostic recommendations, such a strategy may be ideal since serology generally costs less than UBT and may be combined with a blood draw being done for other diagnostic tests. Continuing to offer serology increases options for patients and may provide economic benefits for single-payer health care systems or health maintenance organizations.

**IMPORTANCE** This study compares the performance of serology with urea breath test in the diagnosis of *Helicobacter pylori* in a population-level data set and mimics a head-to-head direct comparison as the study population had both tests performed within 2 weeks of each other. This provides new information supporting the use of serology in a diagnostic algorithm. There are several instances where serology could be preferable to patients to rule out disease, despite some guidelines suggesting serology should not be used.

**KEYWORDS** *Helicobacter pylori*, serology, urea breath test

*Helicobacter pylori* is a Gram-negative bacterium that colonizes the gastric mucosa. Prevalence rates vary worldwide, but in some regions, they are known to exceed 50% (1). In the decades since Warren and Marshall established the role of *H. pylori* in peptic ulcer disease (2), more evidence has emerged of the complex role it plays in human pathophysiology. In addition to its roles in peptic ulcer disease and gastric mucosa-associated lymphoid tissue (MALT) lymphoma (3), it has been associated with immune-driven phenomena such as chronic urticaria (4) and idiopathic thrombocytopenia (5).

The detection of *H. pylori* infection can be a diagnostic challenge. Several available methods each present their own advantages and challenges. While gastroduodenoscopy with biopsy and culture is highly specific, it is invasive and costly compared with

Address correspondence to Miguel Imperial, Miguel.Imperial@cw.bc.ca.

The authors declare no conflict of interest.

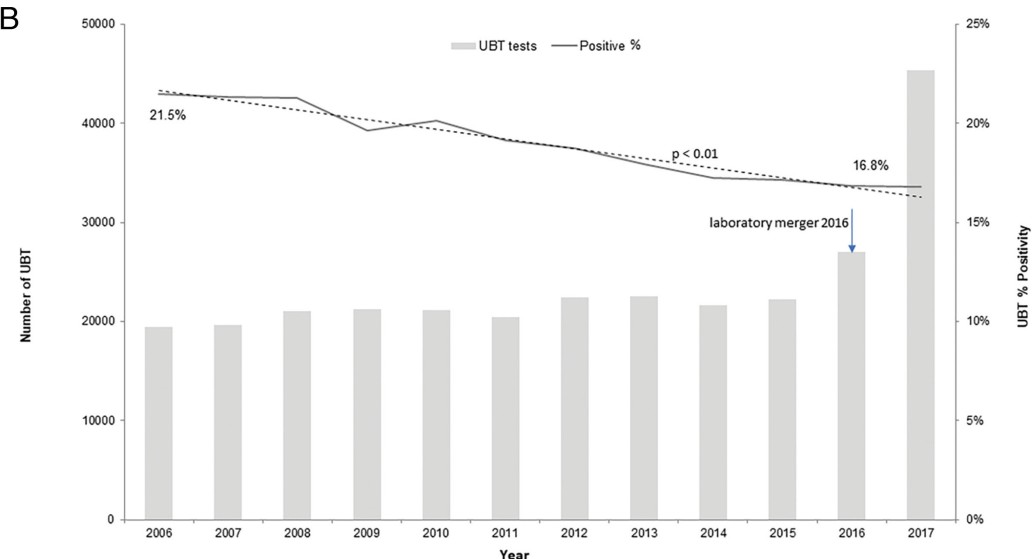

**FIG 1** (A) *H. pylori* serological tests 2006–2017. Only the first result per patient over the study period included in the analysis. Trend line shows a decrease in seropositivity over time, *P* < 0.01. (B) *H. pylori* UBT results 2006–2017. Only the first result per patient over the study period included in the analysis. Trend line shows a decrease in UBT positivity over time, *P* < 0.01. Note that the merger of the two outpatient labs occurred in 2016, resulting in a large increase in test volume starting late 2016. UBT, urea breath test.

non-invasive tests (6). Relatively inexpensive serology for the detection of anti-*H. pylori* IgG is available, but one of the primary disadvantages is it does not differentiate between active and a past resolved infection (7). Non-invasive tests such as the urea breath test (UBT) and the stool antigen test provide evidence of active infection or colonization but are usually more costly than serology and each has inherent challenges with respect to sample collection and analysis (8).

For non-invasive diagnosis, several clinical guidelines, such as those from the American Association of Gastroenterologists (9) and Choosing Wisely Recommendation #14 2020 (10) from the American Society of Clinical Pathology, currently recommend against using serology in favor of the UBT or stool antigen test. In these statements, expert opinion is cited to support this recommendation. The evidence for this opinion is not readily discernible but may be based on a body of conflicting older literature that

**TABLE 1** Characteristics of patients by test ordered, 2006–2017[a]

|  | UBT performed | Serology performed |
|---|---|---|
| n | 395,630 | 586,169 |
| Mean age (years) | 47 | 47 |
| Gender % (M, F, and U) | 39.9, 60.1, 0.0 | 39.9, 59.5, 0.6 |

[a]F, female; M, male; U, undefined; UBT, urea breath test.

used earlier-generation serological assays (11, 12). We are not aware of any published study similar to ours that uses a large population-level data set to compare the performance of a modern commercial serological assay with UBT that can provide more robust contemporary evidence to either support or refute this expert opinion-based recommendation.

In the province of British Columbia, *H. pylori* serological testing is performed using a single centralized laboratory, the British Columbia Centre for Disease Control (BCCDC) reference laboratory, using the IMMULITE platform (Siemens), a solid-phase, two-step chemiluminescent enzyme immunoassay (13). Conversely, the majority of UBTs in BC were performed by two outpatient laboratories, LifeLabs and BC Biomedical Lab, which operated within Metro Vancouver and Vancouver Island and merged in 2016. UBT was performed by a 13C-labeled urea spectrophotometry method (14).

Provincial practice guidelines during the study period recommended serology as the initial diagnostic test for *H. pylori* if there was no known history of infection. UBT was recommended only if there was a known history of *H. pylori* infection. Notwithstanding the practice guidelines, clinicians were not restricted and had full discretion as to which diagnostic test to order. Stool antigen testing was not yet available during most of the study period.

## MATERIALS AND METHODS

During the study period (2006–2017), UBT data were available from LifeLabs and serological data from the BCCDC reference laboratory. Together, these reflect the majority of provincial population data for *H. pylori* serology and UBT for the years 2016–2017. These data included a significant number of patients who underwent both tests within days of each other, allowing for a direct, albeit non-randomized, comparison of the performance of both these methods.

After the study was approved by the University of British Columbia Institutional Review Board, data were extracted from the respective laboratory information systems. Patient data were anonymized using surrogate non-identifiable patient identifiers and the data sets were linked for analysis. To best compare test results for the same patient and same episode of illness, serological and UBT results were compared when the UBT occurred within a 14-day window period after serology was performed. For the various descriptive statistics and analyses, only the respective first test on record was included and the comparator test closest in date was selected.

To assess test performance, we first used UBT as the reference standard and calculated the performance of serology using measures of sensitivity, specificity, positive predictive value, and negative predictive value (NPV). Serology results could be categorized as reactive, non-reactive, and equivocal while UBT results were only either positive or negative. For the purposes of this analysis, all equivocal serology results were excluded. The significance of differences in sets of count values was assessed using

**TABLE 2** *H. pylori* serology and UBT positivity by year[a]

| Year | 2006 | 2007 | 2008 | 2009 | 2010 | 2011 | 2012 | 2013 | 2014 | 2015 | 2016 | 2017 | Average |
|---|---|---|---|---|---|---|---|---|---|---|---|---|---|
| Serology positive (%) | 25.6 | 23.9 | 23.8 | 22.2 | 22.0 | 21.8 | 22.6 | 20.9 | 19.4 | 19.5 | 18.1 | 20.2 | 21.3 |
| UBT positive (%) | 21.0 | 20.5 | 20.4 | 19.0 | 18.9 | 17.9 | 17.4 | 16.8 | 16.0 | 15.9 | 15.6 | 15.6 | 17.5 |

[a]UBT, urea breath test.

**TABLE 3** Serology followed by UBT (within 14 days)

| | UBT[a] | | |
|---|---|---|---|
| **Serology** | **Positive** | **Negative** | **Total** |
| Reactive | 670 | 398 | 1,068 |
| Non-reactive | 24 | 1,520 | 1,544 |
| Total | 694 | 1,918 | 2,612 |

[a]UBT, urea breath test.

Fisher's exact test. Statistical calculations were performed using R Statistical Software (v4.1.2; R Core Team 2021) and Excel 2023 (Microsoft).

## RESULTS

The characteristics for both groups of *H. pylori* testing are shown in Table 1. Mean age and gender were not significantly different between the groups. There was a higher proportion of females undergoing testing by either method.

The number of both unique serology tests and UBT per year (Fig. 1A and B) showed an increasing trend over time. The marked increase in UBTs reported between 2016 and 2017 is due to additional data being available from the merging of the two large outpatient laboratories in the region. Prior to this, only UBT data from one lab (LifeLabs) are represented. The positivity rate for both tests had a statistically significant (regression analysis of trend, $P < 0.01$) decrease over time, represented by the trend lines in Fig. 1A and B, respectively. Positivity rates for each test per year are shown in Table 2.

To assess the concordance between serology and UBT, we looked for the first serological test of each patient in the data set and then looked for the first corresponding UBT, if any. We narrowed this data set to look for patients who had serology first and then UBT subsequently reported within a window of 14 days afterward. A total of 2,612 individuals met these criteria, with the results of their tests summarized in Table 3.

This patient set mimics one that follows the diagnostic algorithm of using serology first as a screen, followed by UBT. However, because the length of the analysis window period (14 days) was arbitrarily selected, we also performed a sensitivity analysis to see if there was any effect of changing this window period. We reversed the original analysis by looking at patients where UBT was performed first followed by serology within 14 days. We also looked at the patient group where both tests were performed on the same exact day, which would most closely mimic a direct head-to-head comparison of diagnostic assays. Lastly, we combined all sets. As seen in Table 4, there was a minimal effect on sensitivity and specificity and NPV.

## DISCUSSION

Our study suggests that in contrast to recommendations from some clinical practice guidelines, serology for *H. pylori* remains a viable diagnostic option. In recent years, numerous diagnostic labs have discontinued offering serology due to these recommendations (10), overlooking instances where serology may be preferred.

**TABLE 4** Sensitivity analysis for time variable in the analysis of serology test performance[a]

| Analysis group | N | Sensitivity (%) | Specificity (%) | Negative predictive value (%) | Positive predictive value (%) |
|---|---|---|---|---|---|
| Serology first and UBT within 14 days after (reference) | 2,612 | 96.5 | 79.2 | 98.4 | 62.7 |
| UBT first and serology within 14 days after | 1,351 | 93.7 | 91.4 | 98.6 | 66.4 |
| UBT and serology same day | 912 | 93.5 | 91.4 | 98.6 | 68.9 |
| Serology and UBT anytime within 28 days of each other | 3,051 | 96.8 | 81.3 | 98.6 | 62.6 |

[a]UBT, urea breath test.

In our jurisdiction, where the observed average active *H. pylori* infection rate based on UBT is 17.5%, using serology (with a sensitivity of 96.5% and specificity of 79.2%) had an NPV of 98.4% compared with the reference standard UBT. This suggests that *H. pylori* serology is sensitive enough for initial screening in populations with low-to-moderate prevalence, effectively ruling out active infection.

For positive serology results, which do not distinguish between active and past infections, the diagnostic algorithm could involve a follow-up UBT or an *H. pylori* stool antigen, as has been suggested by others (15). This approach has some advantages. Many patients undergoing investigations for dyspepsia or peptic ulcers are already having blood work done for other purposes and serology can be easily added to the same blood draw in addition to being performed on high-throughput automated analyzers. In contrast, UBT requires a separate collection involving drinking a liquid and providing a breath sample for analysis. Patients must also cease proton pump inhibitors and any antibiotics for 2 weeks prior to collection. Stool antigen requires a formed (i.e., non-liquid) fecal sample, which is a collection type not often preferred by patients when there are other options (16) and involves additional manual preparation at the lab even with automated methods (17).

In our population, average *H. pylori* seropositivity was only slightly higher at 21.3% compared with 17.5% for UBT, a less than 4% absolute difference. Thus, approximately four of five patients could have *H. pylori* infection ruled out with the initial serology test and only one of five would need follow-up testing with UBT or stool antigen. While a formal economic analysis in our own setting is needed, the lower cost of serology per test suggests the potential cost-effectiveness and efficiency of this approach on a population scale without compromising diagnostic accuracy, particularly for single-payer health systems or health maintenance organizations. Other authors, such as Xie et al. (18), have done such simulations for their populations and concluded that serological screening was the most cost-effective approach. However, a drawback of this two-step approach is the need to recall patients with positive results for UBT (or stool antigen), potentially risking a loss to follow-up.

Another potential drawback is that in higher prevalence populations the NPV of serology is less useful as a rule-out test. However, in our BC population, the data reveal a statistically significant decrease in both *H. pylori* seropositivity and UBT positivity over time, consistent with the trends observed in other regions (19, 20). This finding is likely explained by the hygiene hypothesis (21), especially in the context of a developed nation like Canada. Consequently, as the prevalence of *H. pylori* continues to decrease, serology is expected to become increasingly effective as a rule-out test.

One limitation of our study is that it is retrospective; a prospective, randomized study comparing paired UBT and serology in the same patient compared with control arms tested by UBT and stool antigen alone would have been the ideal design. The retrospective nature of our data means we lack access to the clinical information as to why both tests were performed in close proximity to each other. Such an ordering pattern, particularly where both tests were done on the exact same day, was not endorsed by any clinical guideline and we can only speculate that at least some of these were ordered in error by clinicians. Although the data set mimics a natural experiment, confounding factors affecting our sample of convenience cannot be ruled out. Despite this limitation, the near-population level of the data set reinforces the robustness of our findings as it includes thousands of patients with both *H. pylori* serology and UBT results on the same or nearby days. Sensitivity analysis also showed a minimal impact on results even when considering variations in the analysis time window, supporting the robustness of the findings. Another limitation of this analysis is that while UBT is being used as the reference standard it has been demonstrated that UBT does have false-positive and false-negative results (18). Bosch et al. (15) found that a composite standard of multiple modalities was the best reference standard.

In conclusion, this study offers valuable insights for integrating various testing modalities into a population-level algorithm for *H. pylori* diagnosis. Our data support

that in jurisdictions with similar demographics to ours, a two-step algorithm of serology followed by a more specific test like UBT or stool antigen could balance economic efficiency, diagnostic accuracy, and patient convenience.

## AUTHOR AFFILIATIONS

[1]Department of Pathology and Lab Medicine, University of British Columbia, Vancouver, British Columbia, Canada
[2]BC Women's and Children's Hospital, Vancouver, British Columbia, Canada
[3]Lifelabs, Surrey, British Columbia, Canada
[4]Fraser Health Authority, Surrey, British Columbia, Canada
[5]British Columbia Centre for Disease Control, Vancouver, British Columbia, Canada

## AUTHOR ORCIDs

Miguel Imperial http://orcid.org/0000-0002-4061-7940
Muhammad Morshed http://orcid.org/0000-0003-1841-9320

## AUTHOR CONTRIBUTIONS

Miguel Imperial, Conceptualization, Data curation, Formal analysis, Funding acquisition, Investigation, Methodology, Project administration, Resources, Software, Visualization, Writing – original draft, Writing – review and editing | Kennard Tan, Data curation, Formal analysis, Methodology, Visualization, Writing – review and editing | Chris Fjell, Data curation, Formal analysis, Methodology, Software, Visualization, Writing – review and editing | Yin Chang, Data curation, Formal analysis, Visualization, Writing – review and editing | Mel Krajden, Conceptualization, Methodology, Resources, Supervision, Writing – review and editing | Michael T. Kelly, Conceptualization, Methodology, Project administration, Resources, Supervision, Writing – review and editing | Muhammad Morshed, Conceptualization, Writing – review and editing

## ADDITIONAL FILES

The following material is available online.

### Open Peer Review

**PEER REVIEW HISTORY (review-history.pdf).** An accounting of the reviewer comments and feedback.

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
