## [Reviewer comments · Microbiology Spectrum]

Microbiology Spectrum

Diagnosis of *Helicobacter pylori* Infection - Serology vs. Urea Breath Test

Miguel Imperial, Kennard Tan, Christopher Fjell, Yin Chang, Mel Krajden, Michael Kelly, and Muhammad Morshed

Corresponding Author(s): Miguel Imperial, The University of British Columbia

Review Timeline:

Submission Date:	April 29, 2024
Editorial Decision:	May 17, 2024
Revision Received:	August 16, 2024
Accepted:	August 26, 2024

Editor: Bobby Warren

Reviewer(s): Disclosure of reviewer identity is with reference to reviewer comments included in decision letter(s). The following individuals involved in review of your submission have agreed to reveal their identity: Juan Carlos RODRIGUEZ (Reviewer #1)

Transaction Report:

DOI: <https://doi.org/10.1128/spectrum.01084-24>

Re: Spectrum01084-24 (Diagnosis of Helicobacter pylori Infection - Serology vs. Urea Breath Test)

Dear Dr. Miguel Imperial Imperial:

Thank you for the privilege of reviewing your work. Below you will find my comments, instructions from the Spectrum editorial office, and the reviewer comments.

In your revision, please speak to the importance of your study within the context of previous similar work and why this study is of value to the field.

Revision Guidelines

Sincerely,
Bobby Warren
Editor
Microbiology Spectrum

Reviewer #1 (Comments for the Author):

The diagnostic utility of Helicobacter pylori serology has been repeatedly analyzed for many years and the paper lacks originality. Furthermore, the study, by not including control groups, presents important methodological limitations.

Reviewer #2 (Comments for the Author):

The paper by Imperial et al. describes a retrospective study that compares the results of serology and the urea breath test for *H. pylori* diagnosis. Although the results are as expected and no major new information is added, the study benefits from a very large sample size, confirming previous findings, such as the decreasing prevalence of *H. pylori* over time. The study proposes a two-step approach for diagnosis: using serology initially, and if the result is positive, following up with the urea breath test to distinguish between past and current *H. pylori* infection.

As a suggestion, the decreasing tendency of *H. pylori* in the studied population could be better described and analyzed. Currently, it is only observable in the figure, and the proportion of positives over time is not presented. Additionally, the extent of the decline and whether it is statistically significant is not discussed. It would be beneficial to include a detailed statistical analysis of this trend to enhance the robustness of the findings and provide clearer insights into the changes in *H. pylori* prevalence over time.

Methods

Clearly refer to the study period in the first sentence of the Methods section.

Figure 1A - legends should be more complete. Explain reactive, non-reactive and equivocal.

Minor

H.pylori - add a space, like this: *H. pylori* (use italics)

August 15, 2024

Dear Reviewers,

Thank you for taking the time to review this manuscript.

Reviewer #1.

- Agree that *H.pylori* serology has been studied and reported for years. Its limitations are well known, but there are several studies that do report where it is useful. Surprisingly, recent North American expert opinion recommendations make a broad recommendation to not use serology (addressed in the manuscript) . e.g. from Choosing Wisely: “ Serologic evaluation of patients to determine the presence/absence of Helicobacter pylori (H. pylori) infection is no longer considered clinically useful” without clearly citing an evidence base for such a conclusion. Since the types of serological assays published in the past are together heterogenous and are not that amenable to generalization, this study is unique because we report the results of large number of tests using an FDA approved commercial assay that can be run reliably at high throughput in any modern lab.
- Another well used clinical resource like UpToDate states: that while serology is “inexpensive and non-invasive”, “serologic tests require validation at the local level, which is impractical in routine practice”, and further that guidelines recommend that serologic testing should not be used in low prevalence populations as low accuracy of serology would result in inappropriate treatment”.
- Therefore, the uniqueness of this paper is that it is essentially a large population based “validation” of serology against Urea Breath Test as UpToDate suggests, and provides evidence that serology can still be useful, especially as a rule-out test in a low-prevalence population (contrary to the stamen in UpToDate). The large numbers of tests of this analysis, performed in a heterogenous population makes the results better applicable to other North American jurisdictions.
- Agree that lack of a control group and non-randomized population are limitations. These are acknowledged and remarked upon in the Discussion/Interpretation. However, the inclusion of the sensitivity analysis and varying the analysis time window which changes the study group provides some robustness to the findings, making the concordance less likely due to chance.

Reviewer #2

- We added Table 3 to clearly show the changing positivity % of both UBT and serology over the years, and perform a linear regression statistic to demonstrate a statistically significant decrease in positivity over time. Figure 1A/1B have been revised to better demonstrate the positivity change over time including a trend line. I hope the legend is sufficient as well. We also added a few sentences in the discussion to better point out this trend and its relation to world trends and the hygiene hypothesis.
- Proper spacing to the term *H. pylori* has been added and italicization has been checked for.

Re: Spectrum01084-24R1 (Diagnosis of Helicobacter pylori Infection - Serology vs. Urea Breath Test)

Dear Dr. Miguel Imperial Imperial:

Your manuscript has been accepted, and I am forwarding it to the ASM production staff for publication. Your paper will first be checked to make sure all elements meet the technical requirements. ASM staff will contact you if anything needs to be revised before copyediting and production can begin. Otherwise, you will be notified when your proofs are ready to be viewed.

Sincerely,
Bobby Warren
Editor
Microbiology Spectrum